# Neural Models for Measuring Confidence on Interactive Machine Translation Systems

Ángel Navarro * and Francisco Casacuberta

Research Center of Pattern Recognition and Human Language Technology, Universitat Politècnica de Valencia, 46022 Valencia, Spain; fcn@prhlt.upv.es
* Correspondence: annamar8@prhlt.upv.es

**Abstract:** Reducing the human effort performed with the use of interactive-predictive neural machine translation (IPNMT) systems is one of the main goals in this sub-field of machine translation (MT). Prior works have focused on changing the human–machine interaction method and simplifying the feedback performed. Applying confidence measures (CM) to an IPNMT system helps decrease the number of words that the user has to check through the translation session, reducing the human effort needed, although this supposes losing a few points in the quality of the translations. The effort reduction comes from decreasing the number of words that the translator has to review—it only has to check the ones with a score lower than the threshold set. In this paper, we studied the performance of four confidence measures based on the most used metrics on MT. We trained four recurrent neural network (RNN) models to approximate the scores from the metrics: Bleu, Meteor, Chr-F, and TER. In the experiments, we simulated the user interaction with the system to obtain and compare the quality of the translations generated with the effort reduction. We also compare the performance of the four models between them to see which of them obtains the best results. The results achieved showed a reduction of 48% with a Bleu score of 70 points—a significant effort reduction to translations almost perfect.

**Keywords:** machine translation; confidence measures; neural model; quality estimation; interactive machine translation

## 1. Introduction

Confidence measures (CMs) [1,2] in the machine translation (MT) field estimate the correctness of the translations generated by the system without accessing the ground truth. The confidence estimations generated are compared with a threshold value between zero and one to classify the different elements as correct or incorrect. The CMs can be applied to study the correctness of the system at multiple levels such as words, sentences, or documents. In this article, we investigate the performance of the different estimators at a word level.

Nowadays, the MT systems are not able yet to assess human parity in many tasks [3]. To assure high-quality translations without errors, the companies use professionals to post-edit the translations or generate them with the cooperation of the MT system in interactive environments. In interactive-predictive neural machine translation (IPNMT) systems, the user only has to correct the first error from the translation generated, fixing the correct prefix or all correct subsequences, then the system automatically tries to generate a better translation. This procedure speeds up the work and reduces the human effort, and at the same time, the system generates as output perfect translations. There are domains where error-free translations are not needed, and CMs become helpful. With the use of CMs, the system is classifying in advance the words as correct or incorrect, and the translator only has to check and correct the words classified as incorrect. In a perfect world, the CMs work perfectly and do not classify any word incorrectly, and all the errors found are corrected, generating perfect translations. Currently, CMs perform some mistakes, and the gains in

effort reduction cause a decrease in the quality. For this reason, CMs are mainly used for companies when they need good translations that do not change the understanding of the text.

Confidence measurements are not an exact science, and there is no unique method to calculate them. Over time, the methods used have changed, from the first, based on the study of features extracted from the data, to the more new methods that start to use neural models in their calculation. In this paper, we trained multiple neural models to predict the scores from four of the most used quality metrics of the MT field. We use the model outputs to calculate our confidence measures. Those words that the model determines that suppose a higher quality will also have more probabilities of being correct.

CMs appeared firstly in the speech recognition (SR) field with the same purpose, to obtain a correctness score of the words estimated by the system. Slowly, researchers started to use and study applications of this concept in the machine translation field [4,5]. One of the first techniques used in the MT field extracts multiple features from the dataset to calculate an estimation of the correctness. Specia et al. (2013) [6] developed QuEst, a quality estimation framework that, among others, included 17 features that were summarized as the best in the literature. At the same time, techniques appeared that work with the model information to perform the calculation [6,7], i.e., using the lexicon probabilities from the IBM model 1 or the posterior probability of the neural machine translation model. This technique, where the system extracts the information from the MT model itself, is commonly named glass-box or system-dependent. Similar to Ive et al. (2018) [8], in our project, we trained a neural-based CM, with the main difference that our output is an estimation of the word score obtained from the metric used: Bleu, Meteor, Chr-F, or TER.

To a large extent, much of the research carried out in the CMs field to ease the work performed aim at the post-edition process [9,10]. These applications use CMs to, among others, estimate how much effort each sentence needs [11] or select high-quality segments to publish them without changes [12]. Stherionov et al. (2019) [13] studied the applicability of these techniques with software UI strings from Microsoft products. In addition to the fact that these applications primarily use sentence-level CMs [14], they also require that professional translators fix the whole sentences from the translations with lower confidence. Unlike these works, we aimed at the interactive machine translation field using CMs at the word level. Instead of giving the translator a whole sentence to check, we only ask him to check the first word classified as incorrect from those left.

To rank and assess different CMs [15,16], researchers compare their models using metrics related to the correctness of the scores computed or their classification, e.g., Human-targeted Translation Error Rate (HTER) [17] and Classification Error Rate (CER) [18]. Alva-Manchego et al. (2021) [19] applied the CMs at sentence level in a post-edition environment with real translators to study the time, cost, and quality reduction of using this technique. In addition, we applied CMs at a word level, as Alabau et al. (2013) [20] did in CasMaCat, into an IPNMT environment to study the relationship between the effort reduction in terms of word stroke ratio (WSR) [21] and the decrease in the quality of the translations in terms of BiLingual Evaluation Understudy (Bleu) [22].

We summarized the main contributions of this project to the research on the CM field in two points. First, we develop four new word-level CMs based on the most common metrics used in the MT field, which give a global translation score. To obtain the data for the training, we adopt the technique reward shaping to obtain the word scores of these metrics. The CER scores of these CMs demonstrate their robustness with smooth transitions throughout the threshold range, allowing a better configuration for the next point. Second, we apply these CMs in an IPNMT environment to analyze the relationship between translation quality and effort reduction, where we obtain a decrease in the effort of the 48% with a translation quality score of 70 Bleu points.

Section 2 summarizes the recent projects in the CM field and explains the importance of IPNMT systems and CMs over the years. Our methodology, where we develop four neural models trained on the most used metrics from the MT field and apply them in an

IPNMT environment, is explained in Section 4. Section 5 exposes the experimental setup we followed to develop our project and the different metrics used. These metrics help to compare the classification correctness of each model and the tradeoff between effort reduction and translation quality. The results obtained in Section 6 show that CM scores can be used in IPNMT systems to perform almost perfect translations with a high reduction in the human effort needed. Furthermore, as we set a threshold for the classification system, we can adjust the quality of the translations performed, taking into account that for all the models, a higher quality implies a lower effort reduction. Finally, we discuss in Section 7 that the employment of this technique in a real-world interactive translation workflow could reduce the human effort needed, with the improvement in speed that comes with it.

## 2. Literature Review

The first modern approach to MT dates back to 1949 by Warren Weaver [23]. From then, researchers thought that they would obtain fully automatic high-quality translations in a few years, but that was not the case. In 1966, the Automatic Language Processing Advisory Committee (ALPAC) [24] published a report that concluded that MT was more expensive, less accurate, and slower than human translation and was not likely to reach the high quality sought soon. Even now, with all the improvements made in the MT field, the translations generated are not perfect, and in most cases, the system requires a professional translator to obtain high quality.

In order to obtain high-quality translations, in the beginning, the users had to correct the sentences generated by the machines without any help. Over time, researchers developed new tools to assist humans in the translation process and sped it up, such as the computer-assisted translation (CAT) tools [25]. Between some of the most used CAT tools, we can find translation memories, language search-engines, post-editors, and IPMT systems. Interactive-predictive machine translation (IPMT) focuses on the interaction between the translator and the machine. The MT system generates a hypothesis with the available information, and the user provides feedback to the system to correct it if necessary. Barrachina et al. (2009) [26] firstly introduced this concept.

Researchers continued investigating different techniques to assist the translators in IPMT systems. Projects such as CasMaCat [27] and TransType2 [28] appeared as working environments for translators that incorporated an array of these innovative techniques that were not available in other tools at the time. They combine techniques such as intelligent autocompletion [26], confidence measures [18], prediction length control [29], search and replace, word alignment information [30], and prediction rejection [31]. These projects were one of the first times CMs appeared in CAT tools.

In 2012, Lucia Specia and Radu Soricut started to organize the Quality Estimation shared task at the Workshop on Machine Translation (WMT) [32]. This shared task helps establish the state-of-the-art performance in the field. It also enables researchers to determine new and effective quality indicators and identify alternative machine learning techniques for the problem. Although it started only with sentence-level CMs, nowadays it also uses CMs at the word level.

The investigation in the field increased, and researchers developed new projects and frameworks. The QuEst open-source framework [6] appeared after the first shared task on quality estimation and used the results obtained there. Between others, they implemented the most commonly used features in the task.

The CM field also changed with the apparition of neural models in the MT field. The CM models extracted some of the features used from the statistical MT models, so this paradigm change transformed the techniques and features used in the field. Researchers could now extract new features from the neural MT models and train neural models to predict confidence estimations directly. With these new possibilities, the frameworks OpenKiwi [33] and DeepQuest [8] appeared. Both perform estimations at a sentence and word level, and DeepQuest also allows document level.

Nowadays, CMs are primarily used in post-edition, and researchers study their applicability in this environment. Some projects analyze the relationship between the quality of the translations and the time needed from using CMs in this environment with actual cases [13,19]. The IPMT environment is less used and proposes a new problem as the system performs the estimations simultaneously with the translation generation, so the system does not have the complete translation to calculate the word CMs. Similar to our project, there are still works that continue the investigation in this sub-field [7,34].

## 3. Methodology

This section briefly summarizes the information described in Sections 4 and 5. Those sections explain in more detail how we train and develop the CMs used in the project and the experimental setup we followed.

MT systems use CMs to estimate the correctness of the translations, being able to obtain scores at different levels: document, sentence, and word. The system compares the confidence estimation with a threshold set by researchers to classify the different elements as correct or incorrect. For those elements with a confidence score lower than the threshold, the system classifies them as incorrect. In this project, we worked with word-level CMs and applied them in an IPNMT environment. With this implementation, the user only has to provide feedback about the first word classified as incorrect. By not having to correct those words classified as correct, the system reduces the human effort, but the final translations could have some misclassified words.

The CMs developed in this project are recurrent neural networks (RNN). We trained them with word scores obtained from four of the most used metrics in MT: Bleu, Meteor, Chr-F, and TER.

For the experimentation process, we used the EU corpus. With this corpus, we tested the robustness and the applicability of our CMs. We calculate the percentage of misclassified words they obtain to test their robustness. The dataset was extracted by performing a conventional IPNMT session, saving the correct words and mistakes of the MT system. Then, we applied them in an IPNMT environment to test their applicability by comparing the effort reduction with the translation quality obtained.

Real translators are very costly and take much time for each experiment. For this reason, we opted to simulate the user in the IPNMT environment.

## 4. Confidence Measures

In this project, we studied the performance of four different CMs in the IPNMT field. Each MT field presents a new problem that must be overcome. In the case of IPNMT, the main problem is the computation time of the CMs. In an IPNMT environment, the translator is interactively working with the system to generate the final translations. We need to perform all the calculations in less than 100 ms [35], or the user could feel that the system does not respond instantly and break the workflow. There are several techniques that we can no longer apply due to their high computational time. For this reason, we used neural models that can perform the forward pass to obtain the confidence estimations inside this limitation.

Researchers can apply CMs at different levels: document, sentence, and word. These CMs provide an estimation of correctness. If this value is higher than the threshold set, we classify the element as correct. As our main goal within the field of IPNMT field is to reduce the human effort of translating each sentence, we applied the CMs at the word level. The system classifies all the words from the translation as correct or incorrect, and the translator only has to check the words classified as incorrect. Principally, with this technique, we are reducing the number of words that the user has to type. At the same time, we reduce the cognitive cost that is expended reviewing the words.

Figure 1 describes a conventional prefix-based IPNMT session with word-level CMs. At iteration 0, the system translates the source sentence ($\hat{s}_h$) and classifies all the words ($CM$). Then, at iteration 1, the user has to check the first word classified as incorrect, *"is"*.

In this case, the word is correct, so the user does not perform any changes, and the word is validated. The system corrects translations from left to right, so previous words classified as correct are also validated and saved in the prefix (**p**). At iteration 2, the user checks the next word classified as incorrect, *"a"*. The system has generated in the fourth position a word different from the reference. The user corrects the error, and the system generates a new translation. All the remaining words have been classified as correct, and the translation process finishes. As can be seen in the example, this interactive process no longer assures perfect translations. To achieve this goal, we need accurate CMs that never fail in the classification process.

| | | |
|---|---|---|
| **SOURCE** (x): | | Este documento es un simple instrumento y las instituciones no se hacen responsable de su contenido |
| **REFERENCE** (y): | | Each document is intended for use as a documentation tool and the institutions do not assume any liability for its content |
| **ITER-0** | (**p**) | |
| | ($\hat{s}_h$) | *This document is a simple instrument and the institutions are not responsible for their content* |
| | (*CM*) | OK OK BAD BAD BAD OK OK OK OK OK OK OK OK OK OK |
| **ITER-1** | (**p**) | This document |
| | ($s_t$) | *is a simple instrument and the institutions are not responsible for their content* |
| | (*k*) | is |
| | ($\hat{s}_h$) | *a simple instrument and the institutions are not responsible for their content* |
| | (*CM*) | BAD BAD OK OK OK OK OK OK OK OK OK OK |
| **ITER-2** | (**p**) | This document is |
| | ($s_t$) | *a simple instrument and the institutions are not responsible for their content* |
| | (*k*) | intended |
| | ($\hat{s}_h$) | *simple and the institutions are not responsible for their content* |
| | (*CM*) | OK OK OK OK OK OK OK OK OK OK |
| **FINAL** | (**p**) | This document is intended simple and the institutions are not responsible for their content |

**Figure 1.** Example of a conventional prefix-based IPNMT session with CMs to translate a sentence from Spanish to English. At iteration 1, the user checks a word mistakenly classified as incorrect. At iteration 2, the word "a" is corrected and the system generates a new hypothesis. Non-validated hypotheses are displayed in *italics*, and accepted prefixes are printed in normal font.

To train the CMs, we used four of the most common quality metrics in the MT field: Bleu, Meteor, Chr-F, and TER. These metrics can provide a quality score of the whole sentence using the ground truth translation as a reference. The main problem that we encounter is that to train our neural models, we need the quality scores from the words of the translations, and the metrics do not give this information directly. In order to obtain a quality score for each word of the translation, we use a strategy initially used to overcome the shortcoming of the sparsity of rewards, the reward shaping [36]. Reward shaping helps to distribute a reward between intermediate steps, which we use to distribute our sentence score between all the sentence words. In reward shaping, the intermediate reward for the word at position $t$ of the translation $y$ from the source sentence $x$ is denoted as $r_t(\hat{y}_t, x)$ and is calculated as follows:

$$r_t(\hat{y}_t, x) = R(\hat{y}_{1...t}, x) - R(\hat{y}_{1...t-1}, x) \tag{1}$$

where $R(\hat{y}_{1...t}, x)$ is defined as the metric score of the translation $\hat{y}_{1...t}$ with respect to the source sentence $x$. Note that to use reward shaping, the next equation has to be fulfilled:

$$R(\hat{y}, x) = \sum_{t=1}^{T} r_t(\hat{y}_{1...t}, x) \tag{2}$$

where $T$ is the total length of the translation.

Each of the metrics that we selected in this project take into account different aspects when they calculate the translation scores. In the next section, we see more in detail the four metrics that we selected, and what we can expect from their CMs.

### 4.1. BiLingual Evaluation Understudy (BLEU)

Bleu [22] was one of the first metrics that appeared in the MT field. Like many others, it tries to copy how professionals consider a group of aspects when evaluating translations. Between these aspects, Bleu attempts to reproduce the adequacy, fidelity, and fluency

judgement. The main characteristic of Bleu, and the feature that differentiates it from the others metrics selected, is the use of multiple n-grams orders. Bleu computes the n-gram precision at each level individually, combining them and multiplying the final score by a brevity penalty. Bleu needs this brevity penalty to penalize short sentences with high precision. To calculate the intermediate rewards we wanted this penalty to remain the same so we use the next technique.

When we proceed to calculate the metric score $R(\hat{y}_t, x)$ of the translation, we substitute each word from the segment $y_{t+1...T}$ with a special token *<null>*. In this way, the translation length does not vary when we calculate the reward $R(\hat{y}_t, x)$ for each value of $t$.

We can expect good results from the CM trained with the Bleu word scores. These scores capture the quality of each word pretty well, as they take into account the correctness of the word and its context. In addition, it gives each word the same weight, unlike the metric Chr-F whose use of character n-grams gives more significant weight to the longer words.

### 4.2. Metric for Evaluation of Translation with Explicit ORdering (METEOR)

Meteor [37] appeared to address several weaknesses found in Bleu. Researchers thought that the brevity penalty used in Bleu does not compensate for the nonexistence use of recall. It also opted for a new method to capture the fluency of the translations through the word order. Meteor studies the precision and recall of the unigrams from the translations instead of considering multiple levels of n-grams as does Bleu. To study the fluency of the translation, Meteor calculates the minimum number of chunks that remains the same between the translation and the reference and divides it by the number of unigrams matched. In this case, the length of the translations does not matter as much as in Bleu—Meteor only counts the matched unigrams.

We can expect similar results to those obtained with the Bleu CM. Both metrics consider the correctness of the word using unigrams, but the main difference between them is the process used to capture the context of each word. Meteor captures it through the word order, while Bleu 2, 3, and 4 n-gram matches. This difference is the central aspect that will differentiate the results obtained from both confidence measures.

### 4.3. Chracter n-Gram F-Score (Chr-F)

The character level n-grams have been a critical part of more complex metrics evaluation like MTERATER [38] and BEER [39]. For this reason, Chr-F [40] calculates the F-score based on character n-grams. This metric does not use any penalty related to the length of the translation. The two unique elements it uses to calculate the F-score are precision and recall. The precision works at the n-gram level and the recall at the character n-gram level. Experimentation has proven that this metric correlates very well with human judgment.

Although this metric correlates very well with human judgment at the sentence level, we notice some possible problems when calculating the word quality scores. It uses character-level n-grams in its calculation, giving higher scores to longer words. This aspect is not a problem to calculate the intermediate rewards of the metric, but we think it could cause the confidence measure to face some difficulties during the classification process.

### 4.4. Translation Edit Rate (TER)

Unlike the other metrics, TER [17] is more focused on the correction cost of the translations; it measures the minimum amount of editing that a human would have to perform to change a translation to exactly match the reference sentence. Possible edits include the insertion, deletion, and substitution of single words as well as shifts of word sequences. To calculate TER, we normalized the total number of edits by the length of the reference sentence. In this case, TER does not consider the length of the sentence, and to correctly differentiate the different edition actions, it is necessary to use the special tokens described in the Bleu CM. Otherwise, there are substitutions and word shifts that are not considered correctly.

This metric faces a different problem than the one found in Chr-F. In this case, all the words have the same weight, but TER does not consider their context. This metric is more focused on the correction cost; simplifying it, it only captures whether the word appears, or not, in the same position in the reference. This fact suggests that all the correct words have the same word score, which could be a problem to our CM.

## 5. Experimental Setup

In this section, we explain the different elements involved in the experiments that we carried out. First of all, in Section 5.1, we define the metrics that we use to evaluate the performance of the CMs, the effort reduction, and the quality of the translations. Then, in Section 5.2, we describe the main features of the corpus EU used in this project. In Section 5.3, we explain the architectures selected for the MT model and the CMs. Finally, as we did not use real translators, in Section 5.4 we describe the user simulation method.

### 5.1. System Evaluation

Automatic evaluation of results on MT is a difficult job. Each sentence has multiple possible translations that are good enough to be used, but we use only one as reference to compare the translations generated. By extension, this is a problem that we encounter in our simulation of the IPNMT session. The simulated user will always try to produce the ground truth, although other translations could involve a lower effort. We have to deal with this problem when measuring the user effort and evaluating the quality of the translations, so the results we obtain will be pessimistic.

To evaluate the performance and robustness of our CMs, we use the metric classification error rate (CER) [18]. This metric calculates the number of classification errors of the confidence model divided by the total number of words classified. The main objective of the CMs is to classify the words accurately, using the confidence estimation score and the threshold set. If the model classifies the words correctly, the number of errors will be near zero, as the CER score.

Secondly, we want to analyze the applicability of our CMs in a real-world interactive translation workflow. For this reason, we study the relationship between human effort reduction and the quality of the translations. To evaluate the human effort of the translation process, we selected the metric Word Stroke Ratio (WSR) [21], and to estimate the quality of the translation, we used the metric BiLingual Evaluation Understudy (Bleu) [22].

WSR is computed as the number of word strokes that the user performs to correct a translation, divided by the length of the sentence translated. In this context, we define word stroke as the complete action of correcting one word, but it does not take into account the cost of reading the new suffix provided by the system. Bleu computes a geometric mean of the precision of n-grams multiplied by a factor to penalize short sentences.

### 5.2. Corpora

We carried out all the experiments in the Spanish–English pair of languages of the corpus EU [26], for which statistics are described in Table 1. The corpora contain 214 thousand sentences to train the models, 400 to validate, and 800 to test. We cleaned, lower-cased and tokenized the corpora using the scripts included in the toolkit Moses [41]. Finally, we used the subword subdivision byte pair encoding (BPE) method [42] with a maximum value of 32,000 merges to generate the subwords.

EU is a corpus extracted from the Bulletin of the European Union, which exists in all official languages of the European Union and is publicly available on the Internet. Although we only use the pair of languages Spanish–English, the corpus also contains the pairs German–English and French–English.

**Table 1.** Statistics of the Spanish–English EU corpus. K and M represent thousands and millions, respectively.

|  |  | Es-En | |
|---|---|:---:|:---:|
| | Sentences | 214 K | |
| Training | Average Length | 27 | 24 |
| | Running Words | 6 M | 5 M |
| | Vocabulary | 84 K | 69 K |
| | Sentences | 400 | |
| Dev. | Average Length | 29 | 25 |
| | Running Words | 12 K | 10 K |
| | Sentences | 800 | |
| Test | Average Length | 28 | 25 |
| | Running Words | 23 K | 20 K |

### 5.3. Model Architecture

We used the open-source toolkit NMT-Keras [43] to build the neural models for the MT system and the CM models. We tested the experiments with the recurrent neural network (RNN) architecture. All the systems used Adam [44] as the learning algorithm, with a learning rate of 0.0002. We clipped the $L_2$ norm of the gradient to 5. The batch size was set to 50 and the beam size to 6.

The RNN models used an encoder–decoder architecture with an attention model [45] and LSTM cells [46]. The dimension of the encoder, decoder, attention model, and word embeddings was set to 512. We used a single hidden layer of the encoder and the decoder.

The MT and CM models have the same size and architecture. The main difference between both models lies in the output layer. While the MT model has the usual softmax layer that gives us the probability of each word of the vocabulary, the CM model has a relu layer that provides us with the confidence estimation of the last word sent by input.

### 5.4. User Simulation

One of the problems that we encounter in the IPNMT field at the experimentation process is that having your own team of translators to test the performance of each project and model takes a lot of time per experiment and is expensive. To face this difficulty, we realized the experiments simulating the human translator behavior. The downside of this technique is the fact that we only have one translation reference, and the simulated user does not have in mind all the other possible translations that could involve a lower effort and a higher quality. As we only have one translation in mind, the WSR and Bleu results shown in Section 6 are pessimists.

To simulate the behavior of the translators, we make two assumptions. First, we assume that the CMs never classify a word incorrectly. Second, we suppose that the simulated user is able to correct every word of the translations without any context.

With the first assumption, we force the simulated user to only review those words that the system classifies as incorrect, besides the correctness of the classification. Confidence estimation is not perfect, therefore there will be words misclassified that the user would not correct. However, with these misclassified words, we are testing the performance of the CMs, as we no longer assure perfect translations and study the quality of the end translations.

The second assumption is a consequence of the first one. As we can omit incorrect words due to a mistake in the CMs, the simulated user should be able to rectify the wrong words in every case without any context. As we have the translation reference, when the system gives the user a word to correct, he compares it with the word corresponding to the same position in the reference and rectifies it if necessary. As the simulated user only needs the position of the word to check and correct it, he does not mind the context of the word

in the sentence. Rectifying the words without any context does not cause a problem to our quality metric—Bleu—as it uses matches between segments of 4 n-grams at maximum.

## 6. Results

This section exposes the results obtained with the four CMs trained in this project. Two essential aspects in the CMs field are the study of the robustness and applicability of the models. First, to study the robustness of the model, we test how well they classify the words as correct or incorrect for different threshold values within the range. Second, to analyze the applicability of the models in an IPNMT environment, we studied, for each possible threshold value, the quality of the translations and the human effort needed for that process. With this comparison, we know the relationship between the human effort reduction and the quality of the translations obtained.

### 6.1. Robustness

To study the robustness of the models, we used the metric CER that gives us the percentage of errors that the model has performed during the word classification process. We calculated this score along all the threshold range. Similar to in other metrics, we need a reference to compare and calculate the CER score of our models. As we want to study the behavior of our models in an IPNMT environment, we generated the dataset by translating the test in this environment without using any CM. With this process we obtain a set of correct and incorrect words for each translation. We applied this test set in an IPNMT environment with the four CMs and compare the results with the newly obtained references to generate the final CER scores.

Figure 2 shows an example of the process we followed for each sentence to obtain the correct and incorrect words dataset. As we used a prefix-based IPNMT environment, the simulated user only has to correct the first errors that he finds. In this example, the user found two mistakes: "*written*" in iteration one and "*some*" in iteration two. These two words make up the whole group of incorrect words, while the words from the reference go to the group of correct words. We repeated this process for all the sentences of the test set, creating the whole dataset for the robustness experiment.

Figure 3 shows the results obtained in this experiment for each of the four CMs: Bleu, Meteor, Chr-F, and Ter. We performed the experiment along all the threshold range, but the more significant variation on the CER score happens between the values 0.04 and 0.1. This effect is a cause of the reward shaping used in Section 4 to obtain the word scores for the CMs dataset. The system distributes the total reward of the sentences, for which maximum score is one, between all the words making the word scores compressed in this range of values. With a threshold of 0, the environment behaves as an MT system, classifying all the words as correct. With a threshold of 1, the environment behaves like a conventional IPNMT system, classifying all the words as incorrect. The two extremes of each graphic are the CER scores 0.30 and 0.70.

The transition between the CER scores that obtained the CMs gives us the first indications of how well the CMs will work when applied in IPNMT systems. When there is a considerable change in the CER score in an abridged threshold variation, we will have more trouble applying the CMs in an IPNMT system. There will be less helpful threshold values in these cases, as they tend to achieve similar results in this abridged range of threshold values.

| | | SOURCE (x): | Una versión traducida de un texto. |
|---|---|---|---|
| | | REFERENCE (y): | A translated version of a text. |
| **ITER-0** | (**p**) | | |
| | ($\hat{s}_h$) | | *A written version of a story.* |
| | ( correct words ) | | ( ) |
| | (incorrect words) | | ( ) |
| **ITER-1** | (**p**) | | A |
| | ($s_t$) | | *written version of a story.* |
| | ($k$) | | translated |
| | ($\hat{s}_h$) | | *version of some book.* |
| | ( correct words ) | | ( A ) |
| | (incorrect words) | | ( written ) |
| **ITER-2** | ( **p** ) | | A translated version of |
| | ($s_t$) | | *some book.* |
| | ($k$) | | a |
| | ($\hat{s}_h$) | | *text.* |
| | ( correct words ) | | ( A translated version of ) |
| | (incorrect words) | | ( written some ) |
| **ITER-3** | ( **p** ) | | A translated version of a text. |
| | ($s_t$) | | ( ) |
| | ($k$) | | (#) |
| | ($\hat{s}_h$) | | ( ) |
| | ( correct words ) | | ( A translated version of a text. ) |
| | (incorrect words) | | ( written some ) |
| **FINAL** | ($p \equiv y$) | | A translated version of a text. |

**Figure 2.** Example of a conventional prefix-based IPNMT session without CMs to translate a sentence from Spanish to English. Non-validated hypotheses are displayed in *italics*, and accepted prefixes are printed in normal font. '#' represents the translation validation action. This process obtains a set of correct and incorrect words for each sentence. We used the dataset acquired from this procedure to study the robustness of the CMs.

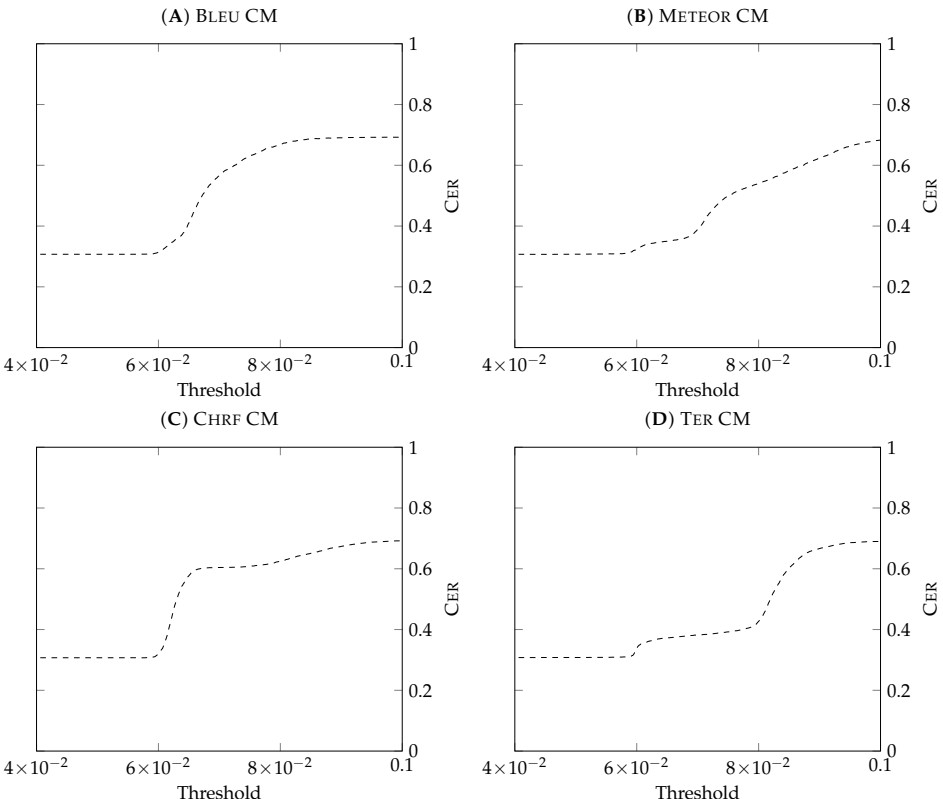

**Figure 3.** CER of the (**A**) Bleu CM, (**B**) Meteor CM, (**C**) Chrf CM, and (**D**) Ter CM. The X-axis represents the threshold values between 0.04 and 0.1, and the Y-axis the CER score.

Considering this effect and the results displayed in Figure 3, where the CER scores of each confidence measure are displayed, we can conclude that the best models are the ones based on Bleu and Meteor metrics. These two models are more likely to obtain better results in the next experiment, where we will test the applicability of the models in an IPNMT environment. We do not discard the models based on Chr-F and TER, but it is true that with the CER scores obtained, these models will have fewer helpful threshold values to select an effort reduction score in the translation process.

### 6.2. Applicability

One of the main objectives of the IPNMT systems is to reduce the human translator effort. This reduction in the human effort also brings an increment indeed in the translation rate. In many cases, companies do not need perfect, error-free translations, and it is in these cases where we can start to study the applicability of our CMs. The CMs are only usable in these cases because they are not perfect yet; when we use them, at the same time that we note a reduction in the human effort, there is also an assumed reduction in the quality of the translations.

To study the applicability of the models, we realized an experiment along all the threshold range, where we simulated an IPNMT translation session with CMs. We annotated the human effort in WSR terms and the translation quality in Bleu terms for each translation. With this information, we can display a graphic along the WSR that gives us the translation quality expected.

Figure 4 displays the results obtained in this experiment for each one of the CMs: Bleu, Meteor, Chrf-F, and Ter. The oversized points in the graphics are the results obtained from all the range of threshold values, and, as can be seen, there are multiple ranges of WSR scores without oversized points. We mentioned this effect in Section 6.1; the quick variation on the CER scores shown in Figure 3 happens because, in these models, there is a large group of words with similar confidence measures, so a bit of change in the threshold suggests that these words change their class. This effect also suggests that the human effort changes significantly due to the classification change of this extensive group of words. We use the results obtained with a threshold of 1 as a baseline, where the system tags all the words as incorrect, and the user has to check them all as per in a conventional IPNMT system. We obtain the maximum WSR, 0.35, and perfect error-free translations in this baseline.

In this experiment, the CMs obtain the best performance when the quality of the translations decreases very slowly with the reduction of human effort. With the WSR, we are only considering the effort that the translators expend in rectifying the wrong words, but at the same time, we are reducing the number of words to check their correctness, reducing the cognitive cost that is involved in their reading and review. Comparing the results obtained from each CM, we can conclude that the model based on the metric Bleu obtained the best results. The transition along the WSR scores is very smooth, the quality of the translations decreases slowly, and there is not a considerable concentration of points in any zone. On the other hand, the other confidence measures decrease the translation quality at a similar velocity and present ranges with a high concentration of points.

After looking at the results obtained in this experiment, we can conclude that the CM with the highest applicability is the model based on the metric Bleu, which obtained, for a translation quality of 70 points of Bleu, a WSR of 0.18 and a 48% reduction in effort. Then, with similar values, we have the models based on Meteor, Ter, and Chr-F, with WSR values of 0.27, 0.28, and 0.29, with an approximate 20% reduction in effort. These last CMs present different transitions for low WSR scores, but in this range of values, they generate translations with a quality almost identical to those obtained with conventional MT systems.

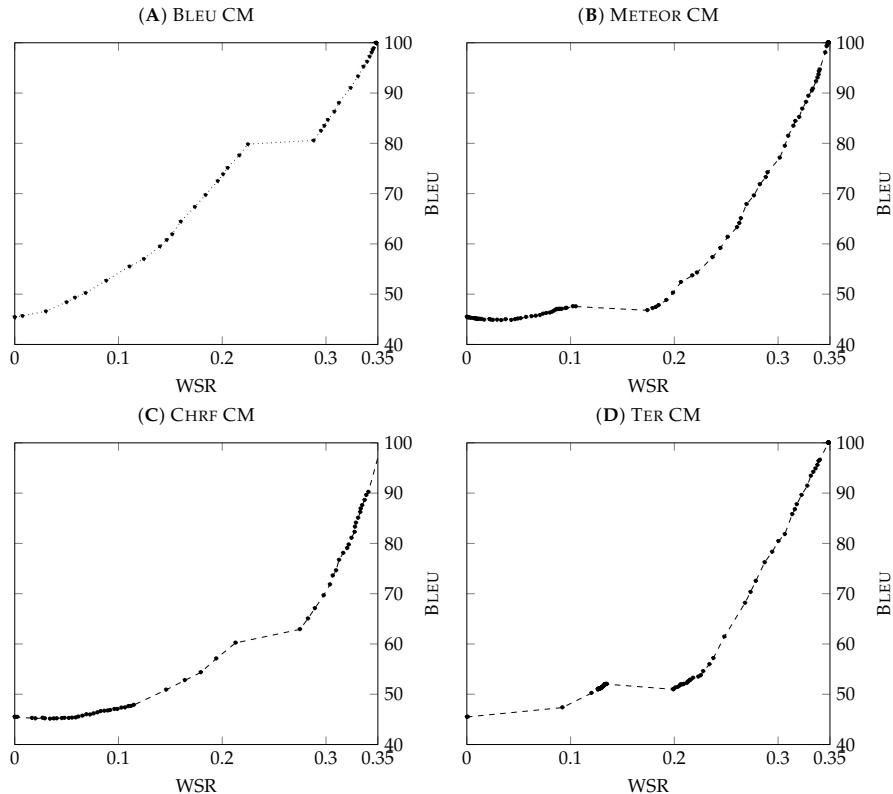

**Figure 4.** WSR compared with the Bleu score of the (**A**) Bleu CM, (**B**) Meteor CM, (**C**) Chrf CM, and (**D**) Ter CM. The X-axis represents the WSR score, and the Y-axis represents the Bleu score obtained with the same threshold value. The larger points are the values obtained from the experiments throughout the threshold range.

Table 2 compares our results obtained for a Bleu score of 70 points with the results obtained in a recent work that uses CMs in an IPNMT environment. Navarro et al. (2021) [7] tried different statistical CMs based on the translation probability of the target word and its alignment probability. They tried CMs based on IBM Model 1 and 2, fast align, and hidden Markov model, and obtained the best results with the fast align model with a WSR reduction of the 19.5% for a Bleu score of 70 points. The Meteor, TER, and Chr-F models from our project obtained similar results, but the CM based in Bleu notably improves upon these results with a WSR reduction of 48%. These results confirm an improvement in the new neural CMs developed in this project applied in IPNMT systems and that there is the possibility of reducing, even more, the human effort with this technique.

**Table 2.** WSR reduction obtained from CMs with a translation quality of 70 Bleu points. IBM-1, IBM-2, fast align, and HMM results from Navarro et al. (2021) [7]. Bleu, Meteor, TER, and Chr-F from our results.

| CM | WSR Red. 70 Bleu Points |
|---|---|
| IBM-1 [7] | 6.6% |
| IBM-2 [7] | 12.6% |
| Fast align [7] | 19.50% |
| HMM [7] | 11.3% |
| Bleu | 48% |
| Meteor | 23% |
| TER | 20% |
| Chr-F | 17% |

## 7. Discussion

Finally, after studying the robustness and the applicability of our CMs in Section 6, we can proceed with the discussion of the results obtained in the experiments. In this section, we define whether the CMs used in this project could be applied in an IPNMT environment and determine which CMs are the most convenient.

When we look at the graphics obtained in Section 6.1, where we studied the CER of the confidence measures, we want to obtain a smooth transition between the two extremes. Between the four CMs that we used, the one based on Meteor presents the smoothest transition. Then we have the CM based on Bleu, which performs a smooth transition through all the threshold range except for the values between 0.65 and 0.7. On the other hand, the CMs based on Chr-F and TER present very low transitions of CER in the vast majority of the graph and perform significant changes very quickly. These quick transitions impair the applicability of the confidence measures, as they suggest that when we compare the WSR versus the Bleu, there are ranges along the WSR without points.

In the second experiment, we studied the applicability by comparing the human effort expended with the quality of the translations generated with the IPNMT environment. This information is helpful and needed when the CMs are applied in a real-world interactive translation workflow, as we need to know which threshold can produce each score of the translation qualities. The model will be more valuable as the more remarkable the human effort reduction, the lower the quality reduction. From the four CMs, the one that better adapts to this definition is the Bleu CM. It has a slow reduction in the quality of the translations, so we obtain a higher effort reduction than in the other models with the same translation quality. The other three CMs tested have similar applicability behaviors; the speed at which the translation quality decreases is almost the same for them.

Between the four CMs that we studied in this project, the one that obtained the best results and has the highest probability of being applied in a real-world workflow is the Bleu CM, due to its smooth transition between the human effort and the translation quality, which allows a better and more secure configuration of the threshold used. Then, we have the Meteor, Chr-F, and Ter CMs that have similar transitions between them.

Finally, after comparing the results that we obtained with our neural CMs based on the most used metrics of the MT field with a previous project that uses statistical CM in the same environment, the improvement in the effort reduction is very notable. Our CMs obtained an effort decrease of 48%, while the higher reduction was about 20% in the previous project.

These results demonstrate that there is still work to do in this field. In future work, now that we have proved the utility of these CMs with simulated users, we will use them with real translators and determine if the results obtained are equivalent to the ones obtained in this project. In addition, in this project, we only worked at a word level, so we will also observe if our CMs are robust and applicable at the sentence level.

**Author Contributions:** Conceptualization Á.N. and F.C.; formal analysis Á.N. and F.C; investigation Á.N.; methodology Á.N.; software Á.N.; supervision F.C.; validation Á.N.; visualization Á.N.; writing—original draft Á.N.; writing—review & editing Á.N. and F.C. All authors have read and agreed to the published version of the manuscript.

**Funding:** This work received funds from the Comunitat Valenciana under project EU-FEDER (*ID-IFEDER*/2018/025), Generalitat Valenciana under project ALMAMATER (*PrometeoII*/2014/030), and Ministerio de Ciencia e Investigación/Agencia Estatal de Investigacion/10.13039/501100011033/ and "FEDER Una manera de hacer Europa" under project MIRANDA-DocTIUM (RTI2018-095645-B-C22).

**Institutional Review Board Statement:** Not applicable.

**Informed Consent Statement:** Not applicable.

**Data Availability Statement:** The EU corpus was acquired and processed in the framework of the TransType 2 (TT2) project, which was supported by the European Commission under reference FP5-IST-2001-32091. The corpus is publicly available on the page https://zenodo.org/record/565309 6#.YZtYN7so-EA (accessed on 15 January 2022).

**Conflicts of Interest:** The authors declare no conflict of interest.

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
