# Peer review of "Neural Models for Measuring Confidence on Interactive Machine Translation Systems"

_applsci, doi:10.3390/app12031100_

Round 1
Reviewer 1 Report
The paper is quite interesting and surely deserves our attention.
However, the English language, despite being 'direct' enough and understandable, needs to be fixed at the level of style - just take a look at the Abstract, "reducing", "reduction", "reduction" are repeated three or more times in a little more than two lines. Ok that this is not a novel, but an academic paper has to be well written, also stylistically.
The Introduction is ok.
This paper has not a Literature Review. It is 'spread' all over the article and it is not clearly presented. The Authors should implement / open a dedicated section, immediately after the Introduction, entitled "Literature Review", listing and commenting the works they used in their paper and providing the readers (also the non-specialized ones) with a succinct, but clear, survey of the current literature available on the topic.
Sections 2 and 3 make for a Methodology section, but it is not always simple to follow them and, despite the 'partition' of section 3 is quite commendable, a specific part of the article, devoted to the Methodology (and entitled "Methodology"), would be very welcomed. It can be directly before the current section 2 (and after the "Literature Review" section I am humbly suggesting) and it can summarize what we can read in the current sections 2 and 3. Then, more specific sections, like the actual ones, with more specific explanations, can be great, but, again, the paper should be read effectively also by a non-specialized audience and, therefore, a good "Literature Review" and a clear "Methodology" section would be very valuable.
Section 4 is ok, the results themselves seem sound and appropriately presented (they need to be double-checked, in any case), my only suggestion would be to expand a little the element of comment and analysis, which is always welcomed and never superfluous - this can be done also in section 5, in that case no expansion of section 4, but expansion of section 5, which could be useful, because the Discussion is too short, it needs a big enhancement, at the level of analytical effort and point-by-point comment.
This paper has not a Conclusion, except for a paragraph in the Discussion. A paper without a Conclusion is relatively 'weird' and not properly acceptable. The Authors, after the Discussion - which, as told, should be enhanced and expanded -, should add a proper Conclusion, highlighting the significance of their paper in their field of studies and summarizing - by 'mirroring' the Abstract and the Introduction - their research goals and how they have realized and achieved them.
All in all, the paper is quite good and surely deserves attention.
What should be done, before publication, is a thorough re-working of the different sections, with reorganization, expansion, and enhancement, and a moderate revision of the English language, as told, mainly at the stylistic level. After that, the paper can surely be reconsidered for publication and would become a valuable piece of research.
Thank you very much.
Author Response
Thanks for your extensive review.
As you suggested, the literature is very spread in the introduction section, so I ended writing the literature review section. Also, I summarized the paper's methodology in a new section. I wrote it in a way that non-specialized audiences could understand.
I also have extended the results section by comparing previous work, which helps to extend the conclusions and show more clearly the highlights of the project.
Reviewer 2 Report
1.The introduction part must addressed the contribution of the article more clear
2.The computational result should be compared with other methods and it should be shown in form of table and has statistically significant test
3.The result part should be related to the introduction parts all contribution in introduction parts that the author addressed must be shown in form of computational result.
Author Response
- I have packed the project's contributions and written a new paragraph that displays them more clearly.
- I have compared the results obtained from the IPNMT implementation with previous work developed in the same environment.
- With the addition of the first point and the comparison added, the result part relates better to the contributions mentioned at the start.
Round 2
Reviewer 1 Report
The paper has been extensively improved and enhanced.
The current format is definitely better and relatively accessible also for a non-specialized audience.
The Methodology section is still too short, and should be expanded, always with the aim of reproducibility.
The overall presentation is effective, although it could be improved further.
Thank you very much.
Reviewer 2 Report
the author fulfill my suggestion, great job